# On the Verification Complexity of Deterministic Nonsmooth Nonconvex Optimization

## Abstract

We study the complexity of deterministic verifiers for nonsmooth nonconvex optimization when interacting with an omnipotent prover and we obtain the first exponential lower bounds for the problem. In the nonsmooth setting, Goldstein stationary points constitute the solution concept recent works have focused on. Lin, Zheng and Jordan (NeurIPS '22) show that even uniform Goldstein stationary points of a nonsmooth nonconvex function can be found efficiently via randomized zeroth-order algorithms, under a Lipschitz condition. As a first step, we show that verification of Goldstein stationarity via determistic algorithms is possible under access to exact queries and first-order oracles. This is done via a natural but novel connection with Carathéodory's theorem. We next show that even verifying uniform Goldstein points is intractable for deterministic zeroth-order algorithms. Therefore, randomization is necessary (and sufficient) for efficiently finding uniform Goldstein stationary points via zeroth-order algorithms. Moreover, for general (nonuniform) Goldstein stationary points, we prove that any deterministic zeroth-order verifier that is restricted to queries in a lattice needs a number of queries that is exponential in the dimension.

## 1 Introduction

Let $f : \mathbb{R}^d \to \mathbb{R}$ be a function that is $L$-Lipschitz, i.e., $|f(x) - f(y)| \le L\|x - y\|_2$, and, lower-bounded, i.e., it satisfies $\inf_{x \in \mathbb{R}^d} f(x) > -\infty$. We study the unconstrained nonsmooth nonconvex optimization problem:

$$\inf_{x \in \mathbb{R}^d} f(x) . \tag{1}$$

This problem, beyond being a fundamental task in the extensive literature of nonsmooth optimization (Clarke, 1990; Makela & Neittaanmaki, 1992; Outrata et al., 2013) (see also Appendix A in Lin et al. (2022)), is highly motivated by modern ML research: standard neural network architectures are nonsmooth (e.g., ReLUs and max-pools). Hence, developing a rigorous theory for nonsmooth optimization is a crucial step towards understanding the optimization landscape of neural networks.

Since the problem of globally minimizing a nonconvex function $f$ (i.e., solving Problem 1) up to a small constant tolerance is computationally intractable (even for smooth and Lipschitz functions) (Nemirovskij & Yudin, 1983; Murty & Kabadi, 1985; Nesterov et al., 2018), researchers have focused on relaxed versions of the problem, by considering alternative solution concepts.

**Clarke Stationary Points.** The problem of finding a Clarke $\epsilon$-stationary point of $f$ is a fundamental relaxed optimization objective. We start with the definitions of generalized gradients (Clarke, 1990) and Clarke stationarity for nonsmooth (and potentially nondifferentiable) functions. The following definition is perhaps the most standard extension of gradients to nonsmooth and nonconvex functions.

**Definition 1** (Generalized Gradient). *The **generalized gradient** of $f$ at $x$ is*

$$\partial f(x) = \operatorname{conv}(u : u = \lim_{k \to \infty} \nabla f(x_k), x_k \to x) ,$$

*namely, the convex hull of all limit points of $\nabla f(x_k)$ over all the sequences $(x_i)$ of differentiable points of $f$ that converge to $x$.*

**Definition 2** ((Near) Clarke (Approximate) Stationary Point). *Given some $\epsilon, \delta \geq 0$ and $f : \mathbb{R}^d \to \mathbb{R}$, we say that $x \in \mathbb{R}^d$ is a **Clarke $\epsilon$-stationary point** of $f$ if*

$$\min\{\|g\|_2 : g \in \partial f(x)\} \leq \epsilon,$$

*and is a $\delta$-**near Clarke $\epsilon$-stationary point** of $f$ if $\|x - x'\|_2 \leq \delta$ and $x'$ is a Clarke $\epsilon$-stationary point of $f$.*

The main results about this solution concept in the nonsmooth nonconvex setting are unfortunately negative: Zhang et al. (2020) shows that there exists no algorithm that can achieve convergence to a Clarke $\epsilon$-stationary point of a general Lipschitz function in a finite number of steps; and Kornowski & Shamir (2021) shows an exponential in the dimension $d$ query lower bound for the problem of finding a $\delta$-near Clarke $\epsilon$-stationary point of a Lipschitz function.

**Goldstein Stationary Points.** Given the above strong negative results, the community focused on a further relaxed yet meaningful notion of stationarity, namely the Goldstein stationary point (or Goldstein subdifferential) (Goldstein, 1977).

**Definition 3** (Goldstein (Approximate) Stationary Point). *Given some $\epsilon, \delta \geq 0$ and $f : \mathbb{R}^d \to \mathbb{R}$, we say that $x \in \mathbb{R}^d$ is a **Goldstein $(\epsilon, \delta)$-stationary point** (or simply $(\epsilon, \delta)$-stationary point) of $f$ if*

$$\min\{\|g\|_2 : g \in \partial_\delta f(x)\} \leq \epsilon,$$

*where $\partial_\delta f(x) = \mathrm{conv}(\cup_{y:\|y-x\|_2 \leq \delta} \partial f(y))$.*

In words, we look at a $\delta$-ball around the point of interest $x$ and for each $y$ in the ball, we take its generalized gradient. In the gradient space, we look at the convex hull of the union of all generalized gradients of points around $x$ and check if there is a point in that set with norm at most $\epsilon$. When $\delta = 0$, we recover the Clarke $\epsilon$-stationary point. We remark that the set $\partial_\delta f(x)$ is called the Goldstein $\delta$-subdifferential of the Lipschitz function $f$ at $x$.[1] For a summary of the extensive prior work on the problem of finding Goldstein stationary points, we refer to Section 1.1

**Uniform Goldstein Stationary Points.** The most natural way to deal with nonsmooth functions is actually (randomized) smoothening, i.e., convert the nonsmooth function into a smooth one and work with tools from smooth nonconvex optimization. This is a fundamental technique in prior work (Jordan et al., 2022; 2023) and gives rise to the important notion of uniform Goldstein stationary points, that we now define.

**Definition 4** (Uniform Goldstein (Approximate) Stationary Point). *Given $\epsilon, \delta \geq 0$ and $f : \mathbb{R}^d \to \mathbb{R}$, we define the function $f_\delta(x) = \mathbb{E}_{u \sim \mathbb{U}}[f(x + \delta u)]$, where $\mathbb{U}$ is the uniform distribution on a unit ball in $\ell_2$-norm, centered at 0. We say that a point $x \in \mathbb{R}^d$ is a **uniform Goldstein $(\epsilon, \delta)$-stationary point** of $f$ (or simply $(\epsilon, \delta)$-uniformly stationary) if $x$ is an $\epsilon$-stationary point of $f_\delta$.*

The next result summarizes the main motivation behind uniform Goldstein points. It essentially says that finding a stationary point of $f_\delta$ implies finding a Goldstein stationary point of $f$ and implies that any bounded, Lipschitz function has at least one Goldstein stationary point (corresponding to a Clarke stationary point of $f_\delta$).

**Lemma 1** (Theorem 3.1 in Lin et al. (2022)). *Suppose that $f$ is $L$-Lipschitz, let $f_\delta(x)$ as in Definition 4 and let $\partial_\delta f$ be the $\delta$-Goldstein subdifferential of $f$, as in Definition 3. Then, we have $\nabla f_\delta(x) \in \partial_\delta f(x)$ for any $x \in \mathbb{R}^d$.*

**Main Questions.** Despite a plethora of recent works on nonsmooth nonconvex optimization (and Goldstein stationarity), a number of fundamental questions have not been answered. First, although recent work has focused on designing algorithms that converge to some Goldstein stationary point, little is known regarding the related problem of *verifying* whether a given point is Goldstein stationary. Note that in the smooth setting, verification (of stationary points) is quite simple since a single query to the gradient oracle suffices to test whether the gradient norm is small. However, after inspecting the quite convoluted definition of Goldstein stationarity, efficient verification of this solution concept seems non-trivial. This gives rise to the following question.

---

[1] It is worth mentioning that Goldstein $(\epsilon, \delta)$-stationarity is a weaker notion than (near) Clarke $\epsilon$-stationarity since any (near) $\epsilon$-stationary point is a Goldstein $(\epsilon, \delta)$-stationary point but not vice versa.

**Q1.** *Is it possible to verify Goldstein stationarity of Lipschitz and bounded functions?*

Regarding the problem of finding Goldstein stationary points, one needs to be careful about the assumptions under which a proposed algorithm works, as they can mask important underlying computational issues. For example, Jordan et al. (2022, Theorem 5.1) shows that although there exists an efficient randomized algorithm for finding a Goldstein stationary point that only uses first-order oracle access to the objective function, no deterministic algorithm can achieve any convergence guarantee, unless a zeroth-order oracle is also provided (in addition to the first-order oracle). Since all of the known zeroth-order algorithms for the problem (Lin et al., 2022; Kornowski & Shamir, 2023) are randomized (and, based on Lemma 1, actually output uniform Goldstein points), the following natural question arises.

**Q2.** *Is there a query efficient zeroth-order deterministic algorithm that finds a uniform Goldstein stationary point of a Lipschitz and bounded function?*

Another related computational issue concerns the impact of finite bit precision. In particular, when the queries to the oracles are constrained to lie on a finite grid, then it is not clear whether the guarantees achieved using exact queries continue to hold, especially since the function is nonsmooth and, therefore, the first-order oracles are not necessarily stable. As a first step, it is, therefore, reasonable to focus on the numerical stability of zeroth-order algorithms.

**Q3.** *How precise queries does a zeroth-order deterministic algorithm need to find a Goldstein stationary point of a Lipschitz and bounded function?*

**Our Contribution.** We make significant progress regarding questions **Q1**, **Q2** and **Q3** using a unified approach based on defining a notion of weak verification (Definition 7), which can be reduced to either (strongly) verifying (Definition 6) or finding (Definition 5) a Goldstein stationary point (see Lemma 2). Therefore, for the problem of (uniform) Goldstein stationarity, *hardness results on weak verification imply hardness on both (strong) verification and finding*, which are, otherwise, two related but not directly comparable problems. We focus on functions that are bounded and Lipschitz.

**Result I (Section 3).** Our first result is a positive answer to **Q1** using a deterministic verifier. We identify a pair of conditions that suffices for this verification procedure: (i) access to exact queries (i.e., points in $\mathbb{R}^d$ with infinite precision) and (ii) access to a first-order oracle $\mathcal{O}_{\nabla f}$ (i.e., given a point $x \in \mathbb{R}^d$, we obtain $\nabla f(x)$). Our algorithm relies on a natural yet novel connection between Goldstein points and Carathéodory's theorem. Hence we answer **Q1** informally as follows.

**Theorem 1** (Informal, see Theorem 4, 5). *There exists a query efficient deterministic verifier for Goldstein stationary points provided access to infinite-precision queries and a first-order oracle.*

**Result II (Section 4).** We show that randomization is actually required for finding *uniform* Goldstein stationary points with query efficient algorithms. As we already mentioned, in Lemma 2 we show that a lower bound on weakly verifying (uniform) Goldstein stationarity implies a lower bound on finding such points. Hence, our main result is an exponential in the dimension lower bound for the problem of weakly verifying uniform Goldstein points. This gives a negative answer to **Q2**.

**Theorem 2** (Informal, see Corollary 1). *There is no query efficient zeroth-order determistic algorithm for weakly verifying (and hence finding) uniform Goldstein stationary points.*

**Result III (Section 5).** We demonstrate that in the nonsmooth setting, weakly verifying a Goldstein stationary point requires sufficiently refined queries to a zeroth-order oracle. In other words, the bit precision of the verifier needs to be sufficiently high.

**Theorem 3** (Informal, see Corollary 2). *Any zeroth-order deterministic algorithm for weakly verifying $(\epsilon, \delta)$-Goldstein stationary points requires queries with coordinate-wise accuracy $O(\epsilon \cdot \delta)$.*

Once more, Lemma 2 can be used to provide the same negative results for the problems of strong verification and of finding (answer to **Q3**). Our result highlights a qualitative difference between smooth optimization (for finding near Clarke approximate stationary points) and nonsmooth optimization (for finding Goldstein approximate stationary points). In particular, for zeroth-order methods in the smooth setting, for instance the work of Vlatakis-Gkaragkounis et al. (2019) shows that

poly$(\epsilon)$ accuracy is sufficient to get $\epsilon$-Clarke stationary points. Hence, the precision of the queries does not need to depend on $\delta$ for the problem of finding $\delta$-near $\epsilon$-approximate Clarke stationary points of smooth functions. In contrast, our result showcases that some dependence on $\delta$ (and $\epsilon$) is necessary for nonsmooth optimization, even when the solution concept is relaxed to $(\epsilon, \delta)$-Goldstein stationarity.

## 1.1 PRIOR WORK

As far as Clarke stationary points are concerned, the work of Zhang et al. (2020) shows that there exists no algorithm that can achieve convergence to a Clarke $\epsilon$-stationary point of a Lipschitz function in a finite number of iterations and the follow-up work of Kornowski & Shamir (2021) shows an exponential in the dimension $d$ query lower bound for the problem of finding a $\delta$-near Clarke $\epsilon$-stationary point of a Lipschitz function. These two results essentially motivated the community to consider Goldstein stationary points as a natural solution concept for nonsmooth nonconvex optimization.

For the problem of finding Goldstein stationary points, there is an extensive literature which we now present. Zhang et al. (2020) gives a randomized algorithm with $O(\delta^{-1}\epsilon^{-3})$ queries to zeroth- and first-order oracles. Davis et al. (2022) focuses on a class of Lipschitz functions and proposed another randomized variant that achieved the same theoretical guarantee. Tian et al. (2022) gives the third randomized variant of Goldstein's subgradient method that achieved the same complexity guarantee. Lin et al. (2022) develops randomized gradient-free methods for minimizing a general Lipschitz function and proved that they yield a complexity bound of $O(d^{1.5}\delta^{-1}\epsilon^{-4})$ in terms of (noisy) 0$^{\text{th}}$ oracles. The works of Jordan et al. (2023; 2022); Kornowski & Shamir (2022) are the closest to our paper. In these works, the authors show a lower bound of $\Omega(d)$ for any deterministic algorithm that has access to both 1st and 0th order oracles, prove that any deterministic algorithm with access only to a 1st order oracle is not able to find an approximate Goldstein stationary point within a finite number of iterations up to sufficiently small constant parameter and tolerance, and, provide a deterministic smoothing approach under the arithmetic circuit model: the resulting smoothness parameter is exponential in a certain parameter $M$ and the method leads to an algorithm that uses $\tilde{O}(M\delta^{-1}\epsilon^{-3})$ queries. Kong & Lewis (2022) gives a deterministic black-box version of the algorithm of Zhang et al. (2020) which achieves, up to a nonconvexity modulus for the objective, a dimension-independent complexity of $O(\delta^{-1}\epsilon^{-4})$. Tian & So (2023) gives computational hardness results for certain first-order approximate stationarity concept for piecewise linear functions. Cutkosky et al. (2023) reduces the complexity of the stochastic rates algorithm of Zhang et al. (2020) (that enjoyed a rate of $O(\delta^{-1}\epsilon^{-4})$) to $O(\delta^{-1}\epsilon^{-3})$ stochastic gradient calls and shows that this rate is optimal. Chen et al. (2023) proposes an efficient stochastic gradient-free method for nonsmooth nonconvex stochastic optimization improving the complexity bound of Lin et al. (2022) regarding the dependence on $\epsilon$ and the Lipschitz constant. Kornowski & Shamir (2023) gives a (randomized) algorithm with optimal dimension-dependence for zero-order nonsmooth nonconvex stochastic optimization with complexity $O(d\delta^{-1}\epsilon^{-3})$.

## 2 SETUP AND PROBLEM DEFINITION

**Black-Box Oracle Access.** We consider oracles that given a function $f(\cdot)$ and a point $x$, return some quantity which conveys local information about the function on that point. We are interested in zeroth-order oracles $\mathcal{O}_f(x) = f(x)$ and first-order oracles $\mathcal{O}_{\nabla f}(x) \in \partial f(x)$. In general, it may be the case that the oracle returns an exact value or an inexact one. In the former case, given $x$, the algorithm observes $\mathcal{O}_f(x)$ and $\mathcal{O}_{\nabla f}(x)$ with infinite precision, while in the latter, it observes a noisy estimate of the exact value. In this work, we focus on exact oracles.

**Finding & Verification Complexity.** We are now ready to present the main algorithmic tasks that we will study in this paper: finding, strong and weak verification using *deterministic* algorithms. For simplicity, we provide the three definitions for Goldstein points (i.e., the problem GOLDSTEIN STATIONARITY) but they naturally extend to other solution concepts, such as uniform Goldstein points (i.e., the problem UNIFORM-GOLDSTEIN STATIONARITY).

Let us start with the most standard notion of finding. In this setting the goal is to design an algorithm that interacts with some oracle and outputs a point that satisfies some desired property, e.g.,

Goldstein stationarity. Randomized or deterministic finding is the task that prior work has focused on (Zhang et al., 2020; Kornowski & Shamir, 2021; Jordan et al., 2022; 2023).

**Definition 5** (Finding Goldstein Points). *Fix $d \in \mathbb{N}$, $\Delta, L > 0$ and let $\mathcal{F}$ be the class of $L$-Lipschitz functions on $\mathbb{R}^d$ with values in $[-\Delta, \Delta]$. (UNIFORM-)GOLDSTEIN STATIONARITY has a **deterministic finding algorithm** if there is a deterministic algorithm $V$ that given $\epsilon, \delta \geq 0$ as input parameters and black-box access to an oracle $\mathcal{O}_f$ for some unknown function $f \in \mathcal{F}$ makes $\ell$ queries to $\mathcal{O}_f$ and outputs a point $x^\star \in \mathbb{R}^d$ such that $x^\star$ is a (uniform) $(\epsilon, \delta)$-stationary point of $f$. We call $\ell = \ell(\epsilon, \delta, d, L, \Delta)$ the deterministic s query complexity of finding for (UNIFORM) GOLDSTEIN STATIONARITY.*

We next turn our attention to strong verification for Goldstein stationarity.

**Definition 6** (Strongly Verifying Goldstein Points). *Fix $d \in \mathbb{N}$, $\Delta, L > 0$ and let $\mathcal{F}$ be the class of $L$-Lipschitz functions on $\mathbb{R}^d$ with values in $[-\Delta, \Delta]$. (UNIFORM-)GOLDSTEIN STATIONARITY has a **deterministic strong certificate of length** $\ell$ if there is a deterministic verifier $V$ that, on input parameters $\epsilon, \delta \geq 0$, point $x^\star \in \mathbb{R}^d$ and black-box access to an oracle $\mathcal{O}_f$ for some unknown $f \in \mathcal{F}$, after making $\ell$ queries to $\mathcal{O}_f$ and interacting for $\ell$ rounds with a (computationally unbounded) prover, outputs a bit $b \in \{0, 1\}$ such that:*

- *(Completeness) If $x^\star$ is a (uniform) $(\frac{\epsilon}{2}, \delta)$-stationary point of $f$, then there exists a prover $P$ such that the output of $V$ is 1.*

- *(Soundness) If $x^\star$ is not a (uniform) $(\epsilon, \delta)$-stationary point of $f$, then for any prover $P$, it holds that the output of $V$ is 0.*

*We call $\ell = \ell(\epsilon, \delta, d, L, \Delta)$ the deterministic query complexity of strongly verifying (UNIFORM) GOLDSTEIN STATIONARITY.*

The main focus of this paper is the weak verification task for Goldstein stationarity.

**Definition 7** (Weakly Verifying Goldstein Points). *Fix $d \in \mathbb{N}$, $\Delta, L > 0$ and let $\mathcal{F}$ be the class of $L$-Lipschitz functions on $\mathbb{R}^d$ with values in $[-\Delta, \Delta]$. (UNIFORM-)GOLDSTEIN STATIONARITY has a **deterministic weak certificate of length** $\ell$ if there is a deterministic verifier $V$ that, given $\epsilon, \delta \geq 0$ as input and black-box access to an oracle $\mathcal{O}_f$ for some unknown function $f \in \mathcal{F}$, after making $\ell$ queries to $\mathcal{O}_f$ and interacting for $\ell$ rounds with a (computationally unbounded) prover, outputs a tuple $(x^\star, b) \in \mathbb{R}^d \times \{0, 1\}$ such that:*

- *(Completeness) If $f$ has at least one (uniform) $(\frac{\epsilon}{2}, \delta)$-stationary point then there exists a prover $P$ such that $V$ outputs $(x^\star, 1)$ for $x^\star$ which is (uniform) $(\epsilon, \delta)$-stationary for $f$.*

- *(Soundness) If $V$ outputs $x^\star$ and $b = 1$, then $x^\star$ is a (uniform) $(\epsilon, \delta)$-stationary point of $f$.*

*We call $\ell = \ell(\epsilon, \delta, d, L, \Delta)$ the deterministic query complexity of weakly verifying (UNIFORM) GOLDSTEIN STATIONARITY.*

We mention that the 'if' condition in the completeness case is always true for functions $f$ that are Lipschitz and lower bounded. We stress that in the above definitions the verifier is independent of the function $f$. We also remark that in the smooth non-convex case, finding stationary points is tractable using deterministic algorithms. Hence both weak and strong verification are tractable too.

One of the main results of this paper (Corollary 1) is that, for the problem of UNIFORM-GOLDSTEIN STATIONARITY, no zeroth-order verifier has deterministic certificates of length sub-exponential in the dimension $d$. To this end, we show that for any interaction sequence of length $\ell = O(2^d)$, any verifier can be fooled in the sense that $x^\star$ is a uniform $(\epsilon, \delta)$-stationary point of $f$ but $V$ says "No" and $x^\star$ is not a uniform $(\epsilon, \delta)$-stationary point of $f$ but $V$ says "Yes".

The next lemma essentially captures the fact that a lower bound for weak verification implies a lower bound for both strong verification and (more interestingly) finding.

**Lemma 2** (Weak Verification Lower Bounds). *The following hold true:*

1. *Assume that* (UNIFORM-)GOLDSTEIN STATIONARITY *has a deterministic strong certificate of length $\ell$. Then it has a deterministic weak certificate of length $\ell$.*

2. *Assume that* (UNIFORM-)GOLDSTEIN STATIONARITY *has a deterministic finding algorithm with query complexity $\ell$. Then it has a deterministic weak certificate of length $\ell$.*

*Proof.* For 1, it suffices to observe that each function in $\mathcal{F}$ (as considered in Definition 7) has some (Uniform) Goldstein stationary point $x^\star$. For weak verification, the prover can suggest $x^\star$ and the strong verifier can check whether it is actually a (Uniform) Goldstein stationary point.

For 2, there is no need for a prover, as the finding algorithm is guaranteed to find a (Uniform) Goldstein stationary point $x^\star$ and the weak verifier may output $(x^\star, 1)$. □

## 3 VERIFICATION VIA CARATHÉODORY'S THEOREM

In this section we investigate conceptual connections between Carathéodory's theorem, a fundamental result in convex geometry and Goldstein stationarity. Let us first recall Carathéodory's theorem:

**Fact 1** (Exact Carathéodory). *Let $S \subseteq \mathbb{R}^d$. If $x \in \mathrm{conv}(S)$, then $x$ is the convex sum of at most $d + 1$ points of $S$.*

We first show that Carathéodory's theorem provides short deterministic strong certificates for the Goldstein stationarity problem given access to exact queries and to a first-order oracle. Due to Lemma 2, this implies the existence of short deterministic weak certificates for the same problem. Note, however, that the existence of a deterministic strong verifier of first-order does not imply the existence of an efficient deterministic finding algorithm of first-order, which is known to be impossible (see Jordan et al. (2022, Theorem 5.1)).

**Theorem 4** (Strong Verification via Carathéodory). *Assume access to exact queries and to a first-order oracle $\mathcal{O}_{\nabla f}$ for $f : \mathbb{R}^d \to \mathbb{R}$, which is Lipschitz and bounded. Then* GOLDSTEIN STATIONARITY *with parameters $\epsilon > 0$ and $\delta \geq 0$ has a deterministic strong certificate of length $d + 1$.*

Let us provide the verification algorithm. Assume that $x^\star$ is given in the input of the strong verification procedure. Moreover, let $y_0, ..., y_d$ be the sequence provided by the computationally unbounded prover. The verifier will output a bit $b$. The algorithm performs the following steps.

---

1. For $i = 0, 1, \ldots, d$, compute $\Delta_i = \|x^\star - y_i\|_2$. If some $\Delta_i > \delta$, output $b = 0$.
2. For $i = 0, 1, \ldots, d$, compute $g_i = \nabla f(y_i)$ by invoking the oracle $\mathcal{O}_{\nabla f}$.
3. Solve the convex program:

$$\min_g \|g\|_2$$
$$\text{subject to } g \in \mathrm{conv}(\{g_0, g_1, \ldots, g_d\}).$$

If the solution $g$ satisfies $\|g\|_2 \leq \epsilon$, output $b = 1$, otherwise output $b = 0$.

---

Let us provide some intuition about the completeness of our verifier. Assume that $x^\star$ is a Goldstein stationary point, i.e., there exists $g'$ with $L_2$ norm at most $\epsilon$ and $g' \in \partial_\delta f(x^\star)$. Hence, by Fact 1 with $S = \partial_\delta f(x^\star)$, we know that $g'$ can be written as a convex combination of at most $d + 1$ points of $\partial_\delta f(x^\star)$. These $d + 1$ points correspond to the vectors $\{g_i\}_{0 \leq i \leq d}$ in the above algorithm. Crucially, the gradients $\{g_i\}_{0 \leq i \leq d}$ can be computed since the points $\{y_i\}_{0 \leq i \leq d}$ are provided by the honest prover. Hence, given this short proof, the algorithm can actually verify that $x^\star$ is a Goldstein stationary point. Soundness of our verifier (against malicious provers) follows by the definition of Goldstein stationarity. For the details about the correctness of the verifier, we refer to Appendix A.

Moreover, we can prove the existence of a deterministic strong certificate for Goldstein stationarity of length that does not depend on the dimension of the domain, using the following approximate version of Carathéodory theorem.

**Fact 2** (Approximate Carathéodory Barman (2015)). *Any point $u$ in a polytope $P \subseteq \mathbb{R}^d$ can be approximated to error $\epsilon$ in $\ell_2$ norm with $O(L^2/\epsilon^2)$ vertices, where $L$ is the radius of the smallest $\ell_2$ ball that contains $P$, in the sense that there exist $v_1, ..., v_k$ vertices of $P$ with $k = O(L^2/\epsilon^2)$ so that $\|v - \frac{1}{k}\sum_i v_i\|_2 \leq \epsilon$.*

Combining a similar argument as the one for Theorem 4 with Fact 2, we acquire the following result, whose proof we defer to Appendix A.

**Theorem 5** (Strong Verification via Approximate Carathéodory). *Assume access to exact queries and to a first-order oracle $\mathcal{O}_{\nabla f}$ for $f : \mathbb{R}^d \to \mathbb{R}$, which is $L$-Lipschitz for some $L > 0$. Then GOLDSTEIN STATIONARITY with parameters $\epsilon > 0$ and $\delta \geq 0$ has a deterministic strong certificate of length $O(L^2/\epsilon^2)$.*

We continue with some comments about the above results. Assuming access to exact queries and first-order oracles, efficient deterministic strong verification is possible; however, we do not have a deterministic finding algorithm (we cannot find the points that describe the desired polytope).

Theorem 4 demonstrates that in the ideal case where we are provided access to exact queries and a first-order oracle for an instance $f$ of the Goldstein stationarity problem, then there are short deterministic certificates. However, for nonconvex and nonsmooth instances, the proposed verifier fails when either the queries are not exact or instead of a first-order oracle we only have access to a zeroth-order oracle.

When the queries are not exact, even if one has access to a first-order oracle, the value $g_i$ computed in Step 2 of the verifier proposed in Theorem 4, might be arbitrarily far from the gradient of the corresponding exact query (which is provided only up to some error), since the gradient is not Lipschitz. When one has only access to a zeroth-order oracle, even provided exact queries, the computation of the gradient is numerically unstable and, therefore, Step 2 of the verification process of Theorem 4 fails once more.

Note that none of the above issues apply to the case where only smooth instances are considered, since differentiation is then robust to small perturbations of the queries and the gradients can be approximated accurately via zeroth-order queries (even non exact ones).

Given the above discussion, the failure of Carathéodory-based deterministic zeroth-order verifiers does not exclude the existence of efficient zeroth-order deterministic algorithms. Having provided an answer to **Q1**, we next focus on **Q2**. Are there query efficient zeroth-order deterministic finding algorithms for the fundamental class of uniform Goldstein points? This is the topic of the upcoming section.

## 4 VERIFICATION LOWER BOUNDS FOR UNIFORM STATIONARITY

In this section, we show that uniform Goldstein stationarity do not *even* have short deterministic weak certificates. This implies hardness (via Lemma 2) for the problem of finding uniform Goldstein stationary points under the constraints of zeroth-order oracle access and deterministic behavior.

Theorem 6 is the main tool towards showing that weak verification of UNIFORM-GOLDSTEIN STATIONARITY has $\exp(d)$ deterministic query complexity. In particular, given a set $S$ of $\exp(d)$ points and a point $x^\star$, we demonstrate a Lipschitz function $f$ (dependent on $S$ and $x^\star$) that vanishes on $S$ and $x^\star$ is not $(\epsilon, \delta)$-uniformly stationary for $f$. Hence, for a verifier that outputs $(x^\star, b = 1)$, considering $f = 0$ would violate the soundness of the verification process.

**Theorem 6** (Construction for Uniform Stationarity). *Let $\epsilon, \delta > 0$ and $x^{(1)}, x^{(2)}, \ldots, x^{(m)}, x^\star \in \mathbb{R}^d$, where $m = 2^{d-3}$. Then, there exists a function $f : \mathbb{R}^d \to [-4\epsilon\delta, 4\epsilon\delta]$ such that $f(x^{(i)}) = 0$ for any $i \in [m]$ and the following are true:*

1. *The function $f$ is Lipschitz continuous with Lipschitz constant $L = 14\,\epsilon$.*

2. *The point $x^\star$ is not $(\epsilon, \delta)$-uniformly stationary for $f$.*

To give some intuition about the construction, the designed function is essentially a linear function with some 'holes'. Each 'hole' is a circular region with center a point $x$ of the input set $S$ and inside

this region the function slowly decreases and eventually vanishes at $x$ (therefore, the radius of the circular region has to be large enough). When the number of holes ($|S|$) is small enough (at most $2^{d-3}$), the norm of the expected gradient of a random point in the ball of radius $\delta$ around $x^\star$ is more than $\epsilon$, because most of the mass is outside of the union of the holes, where the gradient norm is determined by the linear function. In other words, the point $x^\star$ is not $(\epsilon, \delta)$-uniformly stationary.

**Corollary 1.** *Let $\epsilon, \delta > 0$. Any deterministic algorithm for weakly verifying uniform Goldstein points of an unknown, $14\epsilon$-Lipschitz and $4\epsilon\delta$-bounded function over $\mathbb{R}^d$ requires $\Omega(2^d)$ queries to a zeroth-order oracle. Moreover, the same is true for the problems of strongly verifying and finding uniform Goldstein points.*

*Proof.* Suppose there exists such a weak verifier $V$ and let $P$ be the prover specified in the completeness condition of Definition 7. Let $m \le 2^{d-3}$ and $x^{(1)}, x^{(2)}, \ldots, x^{(m)} \in \mathbb{R}^d$ be the queries made by a deterministic weak verifier, suppose that the zeroth-order oracle $\mathcal{O}_f$ always returns 0 on each of them and let $(x^\star, b)$ be the output of $V$ after interacting with $P$, where $x^\star \in \mathbb{R}^d$ and $b \in \{0, 1\}$.

Consider $f_0 \equiv 0$ and $f_1$ the function with the properties specified in Theorem 6 (given the queries $x^{(1)}, x^{(2)}, \ldots, x^{(m)}$ and $x^\star$). Both $f_0$ and $f_1$ are consistent with the answers of the oracle. We can therefore set $\mathcal{O}_f = \mathcal{O}_{f_b}$. Then, $b$ cannot be 0, because $f_0$ is zero and therefore has (Goldstein) stationary points (see completeness condition of Definition 7). Hence, $b = 1$. We arrive to a contradiction, since $x^\star$ is not $(\epsilon, \delta)$-uniformly stationary for $f_1$ as per Theorem 6.

The same result for strong verification and for finding readily follows by Lemma 2. $\square$

We remark that the function constructed in Theorem 6 contains Goldstein stationary points. Extending this lower bound for non-uniform Goldstein points is an interesting open problem.

## 5 Query Precision for Goldstein Stationarity

In this section, we aim to understand **Q3** regarding the query precision required in order to find Goldstein stationary points via query efficient zeroth-order deterministic algorithms. We begin with some notation in order to capture the notion of finite precision.

**Definition 8** (Lattices and Grids). *For $\eta > 0$, the lattice $L_\eta$ of width $\eta$ is the subset of $\mathbb{R}^d$ containing the points $x = (x_1, \ldots, x_d)$ such that $x_j = \eta \cdot k_j$ for some $k_j \in \mathbb{Z}$. For $k \in \mathbb{N}$, the grid $G_{\eta,k}$ is the subset of $\mathbb{R}^d$ containing the points $x = (x_1, \ldots, x_d)$ such that $x_j = \eta \cdot k_j$ for some $k_j \in [-k, k] \cap \mathbb{Z}$.*

The next theorem is the main tool in order to show that any deterministic 0th-order algorithm for weakly verifying an $(\epsilon, \delta)$-Goldstein stationary point of a Lipschitz and bounded function requires queries with coordinate-wise accuracy $O(\epsilon \cdot \delta)$. It states that one can create a Lipschitz and bounded function that vanishes on every point within a lattice of width $\eta$ that contains $x^\star$, such that $x^\star$ is not $(\epsilon, \delta)$-stationary whenever $\eta > 10\epsilon\delta$. We also clarify that for appropriately selected $\epsilon = 1/\text{poly}(d)$, the lattice can contain exponentially many 'interesting' queries, in the sense that they lie within a ball of radius $\delta$ around $x^\star$ (which is the region of interest, according to Definition 3).

**Theorem 7** (Lattice Construction for Goldstein Stationarity). *Let $\delta, \eta > 0$ and $x^\star \in L_\eta$. Then, there is a function $f : \mathbb{R}^d \to [-4\delta, 4\delta]$ such that $f(x) = 0$ for any $x \in L_\eta$ and the following are true:*

1. *The function $f$ is Lipschitz continuous with Lipschitz constant $L = 4$.*

2. *The point $x^\star$ is not $(\frac{\eta}{10\delta}, \delta)$-stationary for $f$.*

*Moreover, if $\eta \le \frac{\delta}{\sqrt{d}}$, we have $|L_\eta \cap \mathbb{B}_d(x^\star, \delta)| \ge 3^d$.*

The main idea behind Theorem 7 is to reduce the construction of $f$ to a geometric question in two dimensions, taking advantage of the symmetry of the lattice. We first focus on the values of the function on some finite grid around $x^\star$, since the function can be identically zero far from $x^\star$ without influencing its stationarity. We construct some piecewise linear function $\tilde{f}$ as described in Figure 1 and acquire $f$ by multiplying $\tilde{f}$ with a function that takes zero values far from $x^\star$.

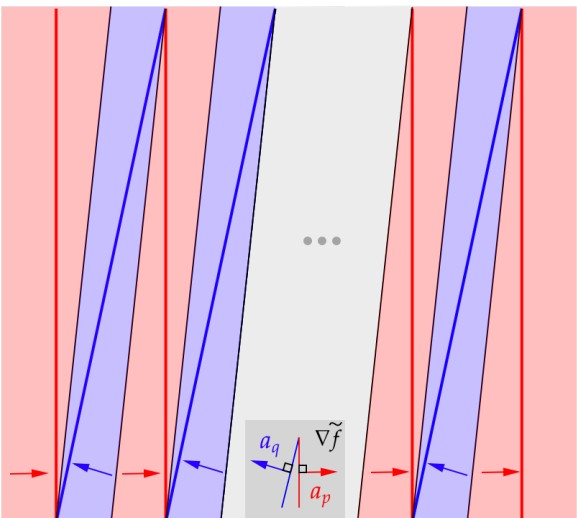

Figure 1: Two-dimensional representation of the constructed piece-wise linear function $\tilde{f}$ projected on the subspace $V_2$ spanned by $(e_1, e_2)$. The function $\tilde{f}$ is constant along any direction orthogonal to $V_2$. The red regions ($R_p^{(\ell)}$) correspond to pieces where the gradient of $\tilde{f}$ is equal to $a_p = e_1$ (red arrow) and the blue regions ($R_q^{(\ell)}$) correspond to pieces where the gradient is equal to $a_q = \frac{-2k}{\sqrt{4k^2+1}}e_1 + \frac{1}{\sqrt{4k^2+1}}e_2$ (blue arrow), for $k = 2\delta/\eta$. The angle between $a_p$ and $a_q$ is bounded away below $\pi$ and hence the minimum norm point in their convex hull is far from the origin. The function $\tilde{f}$ is zero on any point in $\mathbb{R}^d$ whose orthogonal projection on $V_2$ lies on the bold (red or blue) lines.

**Remark 1.** *The function $f$ proposed in Theorem 7, restricted to a ball of rasius $\delta$ around $x^\star$, can be computed by a neural network with $O(\delta/\eta)$ ReLU activation units (ReLU$(t) = \max\{0, t\}$).*

Using Theorem 7, we acquire the following hardness result for the problem of deterministic weak verification of Goldstein stationarity with zeroth-order queries restricted to a lattice.

**Corollary 2.** *Let $\epsilon, \delta > 0$. There is no deterministic algorithm for weakly verifying an $(\epsilon, \delta)$-Goldstein stationary point of an unknown, $4$-Lipschitz and $4\delta$-bounded function, provided access to a zeroth-order oracle with queries restricted to a lattice of width $10 \cdot \epsilon \cdot \delta$. Moreover, the same is true for the problems of strongly verifying and finding Goldstein points.*

*Proof.* The proof is analogous to the one of Corollary 1, but we instead account for non-uniform Goldstein stationarity, based on Theorem 7. The same result for strong verification and for finding readily follows by Lemma 2. □

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

## A    PROOF OF THEOREMS 4 AND 5

For Theorem 4, assume that $x^\star$ is given in the input of the strong verification procedure. Moreover, let $y_0, ..., y_d$ be the sequence provided by the computationally unbounded prover. The verifier will output a bit $b$. The algorithm performs the following steps.

1. For any $i = 0, 1, \ldots, d$, the algorithm computes $\|x^\star - y_i\|_2$. If for some $y_i$ the corresponding computed value is greater than $\delta$, the algorithm outputs $b = 0$ and halts.

2. For any $i = 0, 1, \ldots, d$, the algorithm computes $g_i = \nabla f(y_i)$ by calling the oracle $\mathcal{O}_{\nabla f}$.

3. The algorithm solves the following minimization problem:

$$\min_g \|g\|_2$$

$$\text{subject to } g \in \text{conv}(g_i, i = 0, 1, \ldots, d).$$

   This program is convex and can be solved efficiently (see e.g., De Loera et al. (2018)). If the solution $g$ satisfies $\|g\|_2 \le \epsilon$, the algorithm outputs $b = 1$, otherwise it outputs $b = 0$.

For the promised guarantees we have the following. Recall that $\partial_\delta f(x^\star) = \text{conv}(\cup_{y:\|y-x^\star\|_2 \le \delta} \partial f(y))$.

- If $x^\star$ is $(\epsilon/2, \delta)$−stationary for $f$, then there exists some $g'$ such that $\|g'\| \le \epsilon/2 \le \epsilon$ and $g' \in \partial_\delta f(x^\star)$. By Fact 1, we have that $g' \in \text{conv}(g_i', i = 0, 1, \ldots, d)$, where $g_i' = \nabla f(y_i')$ for some $y_i' \in \mathbb{R}^d$ with $\|x^\star - y_i'\|_2 \le \delta$ (since $g_i'$ are vertices of $\partial_\delta f(x^\star)$ and hence each is a vertex of some $\partial f(y_i') = \nabla f(y_i')$ as specified). Therefore, we can pick $y_i' = y_i$ to get $g_i = g_i'$ (in the second step). In the third step, we will find $g$ such that $\|g\|_2 \le \|g'\|_2 = \epsilon$.

- If $x^\star$ is not $(\epsilon, \delta)$−stationary for $f$, suppose that there are $y_0, y_1, \ldots, y_d \in \mathbb{R}^d$ given by the prover such that the verifier outputs $b = 1$. Then, the algorithm has found $g$ with $\|g\|_2 \le \epsilon$ and $g \in \text{conv}(\nabla f(y_i), i = 0, 1, \ldots, d)$. Moreover, $\|x^\star - y_i\| \le \delta$ for any $i = 0, 1, \ldots, d$. Therefore, $x^\star$ is $(\epsilon, \delta)$-stationary for $f$, which is a contradiction.

For Theorem 5, we use the same algorithm, but the (honest) prover gives, instead, points $y_1, \ldots, y_t$ with $t = O(L^2/\epsilon^2)$, where $(\nabla f(y_i))_i$ correspond to $(v_i)_i$ specified by Fact 2, if we pick $\epsilon$ of Fact 2 as $\epsilon \leftarrow \epsilon/2$ and $u \leftarrow \arg\min_{v \in \partial_\delta f(x^\star)} \|v\|_2$ (we then know that $\|u - \sum_i v_i/k\|_2 \le \epsilon/2$ and, hence, if $x^\star$ is $(\epsilon/2, \delta)$-Goldstein stationary, then $\|\sum_i v_i/k\|_2 \le \epsilon$ (where $\sum_i v_i/k \in \text{conv}((v_i)_i)$). In fact, we could also substitute step 3 with checking whether $\|\sum_i g_i/k\|_2 \le \epsilon$ (and so we can avoid solving the corresponding optimization problem).

## B    PROOF OF THEOREM 6

We first construct a function $\tilde{f} : \mathbb{R}^d \to \mathbb{R}$ as follows.

$$\tilde{f}(x) = \begin{cases} 2\epsilon\, e_1^\top (x - x^\star), & \text{if } \min_{i \in [m]} \|x - x^{(i)}\|_2 > \frac{\delta}{2}, \\ 2\epsilon\, e_1^\top (x - x^\star) \cdot \frac{\min_{i \in [m]} \|x - x^{(i)}\|_2}{\delta/2}, & \text{otherwise.} \end{cases}$$

We first note that the function $\tilde{f}$ is continuous, since $\frac{\min_{i \in [m]} \|x - x^{(i)}\|_2}{\delta/2} = 1$ when $\min_{i \in [m]} \|x - x^{(i)}\|_2 = \frac{\delta}{2}$ and the minimum function is continuous. Moreover, the following are true:

1. $\tilde{f}(x^{(i)}) = 0$ for any $i \in [m]$.

2. For $x \in \mathbb{R}^d$ such that $\min_{i \in [m]} \|x - x^{(i)}\|_2 > \delta/2$, we have $\nabla \tilde{f}(x) = 2\epsilon\, e_1$.

3. For any $x \in \mathbb{R}^d$ we have
   $$|\tilde{f}(x)| \le 2\epsilon \|x - x^\star\|_2.$$

4. For any $x \in \mathbb{R}^d$ on which the gradient of $\tilde{f}$ is defined, we have
   $$\|\nabla \tilde{f}(x)\|_2 \le 2\epsilon \left(1 + \frac{2\|x - x^\star\|_2}{\delta}\right).$$

The above properties can be verified directly, using additionally the fact that $\nabla\|x\|_2 = \frac{x}{\|x\|_2}$. In order to bound the range of the final construction, we consider the following helper function $h : \mathbb{R}^d \to [0,1]$ (radial ramp):

$$
h(x) = \begin{cases} 1, & \text{if } x \in \mathbb{B}_d(x^\star, \delta), \\ 0, & \text{if } x \in \mathbb{R}^d \setminus \mathbb{B}_d(x^\star, 2\delta), \\ 2 - \frac{\|x - x^\star\|_2}{\delta}, & \text{otherwise}. \end{cases}
$$

We then let $f(x) = \tilde{f}(x) \cdot h(x)$. Since $h$ and $\tilde{f}$ are both continuous, $f$ is also continuous. Moreover, the following are true:

1. $f(x^{(i)}) = \tilde{f}(x^{(i)}) \cdot h(x^{(i)}) = 0$ for any $i \in [m]$.

2. For $x \in \mathbb{R}^d$ such that $\min_{i \in [m]} \|x - x^{(i)}\|_2 > \delta/2$ and $\|x - x^\star\|_2 < \delta$, we have $\nabla f(x) = 2\epsilon\, e_1$, since $f(x) = \tilde{f}(x)$ for any $x$ within $\mathbb{B}_d(x^\star, \delta)$.

3. For any $x \in \mathbb{R}^d$ we have

$$
|f(x)| = |\tilde{f}(x)| \cdot |h(x)| \leq 2\epsilon\|x - x^\star\|_2 \cdot \mathbb{1}\{\|x - x^\star\|_2 \leq 2\delta\} \leq 4\epsilon\delta.
$$

4. For any $x \in \mathbb{R}^d$ on which the gradient of $f$ is defined we have

$$
\nabla f(x) = \nabla\tilde{f}(x)\cdot h(x) + \tilde{f}(x)\cdot\nabla h(x) = \begin{cases} \nabla\tilde{f}(x), & \text{if } x \in \mathbb{B}_d(x^\star, \delta), \\ 0, & \text{if } x \in \mathbb{R}^d \setminus \mathbb{B}_d(x^\star, 2\delta), \text{ else} \\ \nabla\tilde{f}(x) \cdot \left(2 - \frac{\|x-x^\star\|_2}{\delta}\right) - \tilde{f}(x) \cdot \frac{x-x^\star}{\delta \cdot \|x-x^\star\|_2}. \end{cases}
$$

   Therefore we get that

$$
\|\nabla f(x)\|_2 \leq 2\epsilon(1+4) \cdot 1 + 4\epsilon = 14\epsilon.
$$

The Lipschitz property follows from the fact that $f$ is continuous and consists of a finite number of pieces in each of which it is differentiable and with bounded gradient.

It remains to show that $x^\star$ is not $(\epsilon, \delta)$-uniformly stationary for $f$. Let us define the uniform smoothening

$$
f_\delta(x) = \mathbb{E}_{u \sim \mathbb{U}(\mathbb{B}_d(0,1))}[f(x + \delta u)],
$$

which is everywhere differentiable Bertsekas (1973)(Proposition 2.4). Hence, $\nabla f_\delta(x)$ exists for any $x \in \mathbb{R}^d$ and is equal to $\nabla f_\delta(x) = \mathbb{E}_{u \sim \mathbb{U}(\mathbb{B}_d(0,1))}[\nabla f(x + \delta u)]$ (see Theorem 3.1 in Lin et al. (2022)).

We next argue about $x^\star$. It holds that

$$
\begin{aligned}
\|\nabla_\delta f(x^\star)\|_2 = \|\mathbb{E}_{x \sim \mathbb{B}_d(x^\star, \delta)}[\nabla f(x)]\|_2 &\geq \frac{\mathrm{V}_d(\delta) - m\mathrm{V}_d(\delta/2)}{\mathrm{V}_d(\delta)} \cdot 2\epsilon - \frac{m\mathrm{V}_d(\delta/2)}{\mathrm{V}_d(\delta)} \cdot 4\epsilon \\
&= \frac{\mathrm{V}_d(\delta) - 3m\mathrm{V}_d(\delta/2)}{\mathrm{V}_d(\delta)} \cdot 2\epsilon \\
&= \frac{\delta^d - 3m(\frac{\delta}{2})^d}{\delta^d} \cdot 2\epsilon > \epsilon.
\end{aligned}
$$

This concludes the proof. In the above, we used the fact that the volume of the $d$-dimensional ball with radius $r$ is $\mathrm{V}_d(r) = \mathrm{V}_d(1) \cdot r^d$, where $\mathrm{V}_d(1) = \frac{\pi^{d/2}}{\Gamma(\frac{d}{2}+1)}$.

## C  PROOF OF THEOREM 7

Suppose, without loss of generality, that $x^\star = 0$. Let $k \in \mathbb{N}$ to be specified later. We will first construct a function $\tilde{f}$ that is Lipschitz and $\tilde{f}(x) = 0$ for any $x \in \mathrm{G}_{\eta,k}$, for which $x^\star$ is not a Goldstein stationary point (for some choice of the stationarity parameters). We will then use $\tilde{f}$ to construct $f$ as desired.

We let $a_p = e_1$, $a_q = \frac{-2k}{\sqrt{4k^2+1}}e_1 + \frac{1}{\sqrt{4k^2+1}}e_2$, $b_p^{(\ell)} = -\ell\eta$ for $\ell \in [-k, k] \cap \mathbb{Z}$ and $b_q^{(\ell)} = \frac{(2\ell+1)k\eta}{\sqrt{4k^2+1}}$ for $\ell \in [-k, k-1] \cap \mathbb{Z}$. We define the function $\tilde{f}$ as follows:

$$\tilde{f}(x) = \begin{cases} a_p^\top x + b_p^{(-k)}, & \text{if } (a_q - a_p)^\top x \geq b_p^{(-k)} - b_q^{(-k)}, \\ a_q^\top x + b_q^{(\ell)}, & \text{if } (a_q - a_p)^\top x \in [b_p^{(\ell+1)} - b_q^{(\ell)}, b_p^{(\ell)} - b_q^{(\ell)}), \text{ for } \ell \in [-k, k-1] \cap \mathbb{Z}, \\ a_p^\top x + b_p^{(\ell)}, & \text{if } (a_q - a_p)^\top x \in [b_p^{(\ell)} - b_q^{(\ell)}, b_p^{(\ell)} - b_q^{(\ell-1)}), \text{ for } \ell \in [-k+1, k-1] \cap \mathbb{Z}, \\ a_p^\top x + b_p^{(k)}, & \text{if } (a_q - a_p)^\top x < b_p^{(k)} - b_q^{(k-1)}. \end{cases}$$

The function $\tilde{f}$ is piece-wise linear and continuous by construction. In particular, we have that $b_p^{(\ell+1)} - b_q^{(\ell)} < b_p^{(\ell)} - b_q^{(\ell)} < b_p^{(\ell)} - b_q^{(\ell-1)}$ and the only candidate points of discontinuity for $\tilde{f}$ are the points $x$ such that $(a_q - a_p)^\top x = b_p^{(\ell)} - b_q^{(\ell)}$ or $(a_q - a_p)^\top x = b_p^{(\ell+1)} - b_q^{(\ell)}$ for some $\ell \in [-k, k-1] \cap \mathbb{Z}$. However, $(a_q - a_p)^\top x = b_p^{(\ell)} - b_q^{(\ell)}$ implies $a_q^\top x + b_q^{(\ell)} = a_p^\top x + b_p^{(\ell)}$ and $(a_q - a_p)^\top x = b_p^{(\ell+1)} - b_q^{(\ell)}$ implies $a_q^\top x + b_q^{(\ell)} = a_p^\top x + b_p^{(\ell+1)}$. Therefore, $\tilde{f}$ is continuous. Since, additionally, in each linear piece of $\tilde{f}$, the gradient norm is at most $\max\{\|a_p\|_2, \|a_q\|_2\} = 1$, we have that $\tilde{f}$ is Lipschitz continuous with Lipschitz constant 1.

Consider the hyperplanes $(p_\ell)_\ell$ and $(q_\ell)_\ell$ in $\mathbb{R}^d$ as follows:

$$p_\ell = \{x : a_p^\top x + b_p^{(\ell)} = 0\} = \{x \in \mathbb{R}^d : x_1 = \ell\eta\}, \text{ for } \ell \in [-k, k] \cap \mathbb{Z},$$

$$q_\ell = \{x : a_q^\top x + b_q^{(\ell)} = 0\} = \{x \in \mathbb{R}^d : x_2 = 2kx_1 - (2\ell+1)k\eta\}, \text{ for } \ell \in [-k, k-1] \cap \mathbb{Z}.$$

Moreover, let $R_p^{(\ell)}$ (resp. $R_q^{(\ell)}$) be the subset of $\mathbb{R}^d$ containing $x$ such that $\tilde{f}(x) = a_p^\top x + b_p^{(\ell)}$ (resp. $\tilde{f}(x) = a_q^\top x + b_q^{(\ell)}$). Note that each $R_p^{(\ell)}, R_q^{(\ell)}$ is an intersection of at most two halfspaces. Then, the following are true:

$$\tilde{f}(x) = 0, \text{ for any } x \in \bigcup_\ell \left(p_\ell \cap R_p^{(\ell)}\right) \cup \left(q_\ell \cap R_q^{(\ell)}\right), \tag{2}$$

$$G_{\eta,k} \subseteq \bigcup_\ell \left(p_\ell \cap R_p^{(\ell)}\right). \tag{3}$$

Equation equation 2 follows from the definition of $p_\ell$ and $q_\ell$, while Equation equation 3 follows from simple geometric arguments in two dimensions (see Figure 1; the bold red lines correspond to the hyperplanes $(p_\ell)_\ell$ and the bold blue lines correspond to the hyperplanes $(q_\ell)_\ell$. Note that the orthogonal projection of the grid $G_{\eta,k}$ on $V_2$ lies within the union of the bold red lines (Eq. 3).

Finally, for any $\delta' > 0$, we have that $\partial_{\delta'}\tilde{f}(x^\star) = \text{conv}(a_p, a_q)$ and, hence, the following is true whenever $k \geq 1$ (recall that $\tilde{f}$ depends on the choice of $k$):

$$\min_{g \in \partial_{\delta'}\tilde{f}(x^\star)} \|g\|_2 = \left\|\frac{1}{2}a_p + \frac{1}{2}a_q\right\|_2 \in \left[\frac{1}{5k}, \frac{1}{3k}\right].$$

The function $\tilde{f}$ does not have all of the desired properties. In particular, while it takes zero values on $G_{\eta,k}$, it does not take zero values on all of the points in $L_\eta$ and, also, its values are not bounded. In order to resolve these issues, we consider the following helper function $h : \mathbb{R}^d \to [0, 1]$ (cylindrical ramp)

$$h(x) = \begin{cases} 1, & \text{if } (x_1, x_2) \in \mathbb{B}_2(0, \delta), \\ 0, & \text{if } (x_1, x_2) \in \mathbb{R}^d \setminus \mathbb{B}_2(0, 2\delta), \\ 2 - \frac{\sqrt{x_1^2 + x_2^2}}{\delta}, & \text{otherwise}. \end{cases}$$

Finally, we let $f(x) = h(x) \cdot \tilde{f}(x)$. We pick $k = \frac{2\delta}{\eta}$, so that $\{x : (x_1, x_2) \in \mathbb{B}_2(0, 2\delta)\} \subseteq G_{\eta,k}$ and therefore $f(x) = 0$ for any $x \in L_\eta$. For any $x$ such that $(x_1, x_2) \in \mathbb{B}_2(0, \delta)$, we have that $|f(x)| = |\tilde{f}(x)| \leq \delta$ (since $\tilde{f}(0) = 0$ and $\tilde{f}$ is 1-Lipschitz). For $x$ such that $(x_1, x_2) \in \mathbb{B}_2(0, 2\delta) \setminus \mathbb{B}_2(0, \delta)$ we have that $|f(x)| = |h(x)| \cdot |\tilde{f}(x)| \leq 2 \cdot 2\delta = 4\delta$. Hence, $f$ is zero on $L_\eta$ and bounded on $\mathbb{R}^d$.

The function $f$ is continuous, since $h$ and $\tilde{f}$ are continuous. Moreover, the Lipschitz constant of $f$ is 4, which can be shown as follows. Let $x$ such that $f$ is differentiable on $x$ and also $(x_1, x_2) \in \mathbb{B}_2(0, 2\delta) \setminus \mathbb{B}_2(0, \delta)$ (everywhere else $f$ is either identical to $\tilde{f}$ or to 0). Then we have

$$\|\nabla f(x)\|_2 \leq \|\nabla h(x)\tilde{f}(x)\|_2 + \|h(x)\nabla\tilde{f}(x)\|_2$$
$$\leq \frac{1}{\delta} \cdot 2\delta + 2 \cdot 1 = 4. \qquad \text{(since } \|\nabla h(x)\|_2 = 1/\delta\text{)}$$

We show that $x^\star$ is not $(\frac{\eta}{10\delta}, \delta)$-stationary for $f$. We have that

$$\min_{g \in \partial_\delta f(x^\star)} \|g\|_2 = \min_{g \in \partial_\delta \tilde{f}(x^\star)} \|g\|_2 \geq \frac{1}{5k} = \frac{\eta}{10\delta}.$$

We will now show that $\tilde{f}$ can be expressed as a ReLU network with $O(k) = O(\delta/\eta)$ neurons. In particular, if we set $a_p^\top x + b_p^{(\ell)} = \tilde{f}_p^{(\ell)}(x)$ and $a_q^\top x + b_p^{(-k)} = \tilde{f}_q^{(\ell)}(x)$, it is not hard to see that

$$\tilde{f} = \max\left\{\dots \min\left\{\max\left\{\min\left\{\tilde{f}_p^{(-k)}, \tilde{f}_q^{(-k)}\right\}, \tilde{f}_p^{(-k+1)}\right\}, \tilde{f}_q^{(-k+1)}\right\} \dots, \tilde{f}_p^{(k)}\right\}$$

By observing, additionally, that $\max\{\alpha, \beta\} = \text{ReLU}(\alpha - \beta) + \beta = -\min\{-\alpha, -\beta\}$ for any $\alpha, \beta \in \mathbb{R}$, we conclude that $\tilde{f}$ can be expressed as a ReLU network with $\Theta(k)$ nonlinear units and depth $\Theta(k)$.

Finally, we observe that for $x \in L_\eta$ we have $\|x\|_2 = \eta\sqrt{\sum_{j=1}^d {k_j}^2}$ and if $k_j \in \{-1, 0, 1\}$, then $\|x\|_2 \leq \eta\sqrt{d}$. Therefore, if $\eta \leq \delta/\sqrt{d}$, then $|L_\eta \cap \mathbb{B}_d(0, \delta)| \geq 3^d$.

