# OpenReview forum: "On the Verification Complexity of Deterministic Nonsmooth Nonconvex Optimization"
_ICLR.cc/2024/Conference — Submitted to ICLR 2024_

### Official Review · Reviewer_FB68 · 2023-10-30

**Soundness:** 3 good
**Presentation:** 2 fair
**Contribution:** 2 fair
**Rating:** 3
**Confidence:** 3

**Summary:**

This paper focuses on the topic of nonsmooth nonconvex optimization, with a specific focus on the complexity of deterministically verifying whether a given point is a (uniform) Goldstein stationary point given a certificate and can interact with a computationally unbounded prover. The authors developed a query efficient deterministic verifier given access to an infinitely-precise first-order oracle, and showed that there is an exponential lower bound on query complexity if the verifier only have access to a zeroth-order oracle. Moreover, the authors showed that the bit precision of the verifier needs to be sufficiently high if they only have access to a zeroth-order oracle.

**Strengths:**

The authors provided an intriguing approach on investigating the complexity of nonsmooth nonconvex optimization. This perspective, which has not been widely discussed in prior works, offers a novel and potentially valuable insight both theoretically and in practical situations.

**Weaknesses:**

1. Some technical details of the paper are very confusing to me.

(1). In Definition 6 and 7, the author mentioned that the prover has unbounded computational power. This property, however, is never used in their proofs.

(2). The interaction scheme between the verifier and the prover is confusing. Since the verifier is deterministic, any multi-round interaction protocol can be transformed into a single round interaction protocol with the same amount of communication complexity, i.e., the prover can send everything to the verifier at the very beginning of the protocol, since they can simulate the deterministic actions of the verifier.

2. This paper only discussed the deterministic verification complexity of (uniform) Goldstein stationary point. However, from my vantage point, it appears that a randomized protocol may be a more general and intuitive choice, given that randomness is not a scarce resource for machine learning algorithms. Also, in a randomized protocol, the interaction scheme described in Definition 6 and 7 makes more sense, since in this case a multi-round interaction is indeed much powerful than a one-round interaction. The authors should provide further clarification and justification for their choice to focus on the deterministic model.

3. (minor comment) The proof technique of this work is intuitive but also arguably, relatively straightforward.

**Questions:**

Corresponding to my comments above, I have the following two questions:

1. Why did the authors introduced an interactive protocol in their deterministic setting?

2. What is the randomized verification complexity, or what is the motivation of considering a deterministic setting instead of the randomized setting?

---

> ### Author Response · Authors · 2023-11-22
>
> We thank the anonymous reviewer for their comments.
>
> In our paper we provide upper bounds for the verification complexity when one is provided with exact queries and oracles (Theorem 4). For this result, we require access to an omnipotent prover, so that the points $y_0,y_1,\dots,y_d$ for Theorem 4 can be generated by the prover.
>
> We focus on deterministic verifiers in order to provide lower bounds for the verification complexity (Theorems 2,3), since in the randomized setting, upper bounds are known. Our work is the first to provide an exponential separation between deterministic and randomized non-smooth non-convex optimization (Theorem 2).
>
> Regarding the interactive protocol, we agree that it is equivalent to a single round of interaction, but this is only a presentational issue and we will consider simplifying the model accordingly for future versions of our paper.

---

> > ### Comment · Reviewer_FB68 · 2023-11-23
> >
> > I would like to thank the authors for answering my questions. I agree with the authors that the interactive protocol does not have technical issues but only presentation issues. However, I think those presentation issues may require major modifications due to their departure from usual presentations in previous papers on interactive proof systems. The assumption on the unbounded computational power is another example. In a one-round interactive proof system, say the complexity class NP, it is usually not given as an explicit assumption that the prover has omnipotent power.
> >
> > In general, I think this work is nice and definitely publishable. Nevertheless, I think the current version without major modifications falls short of meeting the standard of ICLR.

---

> ### Author Response · Authors · 2023-11-23
>
> Thank you for your feedback! We commit to improving the presentation of the paper.
>
> We respectfully disagree with the reviewer on the comment about the power of the prover though. In the definition of NP there is no restriction on the computational power of the prover, the only restriction is that prover outputs a polynomial size proof. Otherwise, the time that it takes for the prover to find the proof is not relevant to the definition of NP. In this sense the prover in the definition of NP has the flexibility of using omnipotent power. The prover might not need to use all this power in every single instance but it is definitely allowed to use omnipotent power if needed. Exactly the same happens in our case! We do not restrict the power of the prover, exactly the same as in NP, and only in this sense our prover has omnipotent power.
>
> In general, we follow the classical definitions from  complexity theory throughout the paper, we do not introduce any new notions that's why we do not emphasize or explain these notions carefully, these are classical definitions not part of our contribution.
>
> We will try to expand the explanation for people that might not be familiar with these definitions but (1) we do believe that this verification problem gives a lot of insights for non-smooth optimization which are relevant to the optimization community, and (2) given this we understand that for a community like ICLR explaining these concepts in the paper has some value but due to the space limitations is tricky to choose how much space to spend on explaining definitions that are not part of our contribution vs focusing on our results. Would it help to add additional explanation of these concepts in the appendix?
>
> Any feedback/advice on that would be really helpful!

---

### Official Review · Reviewer_hBbf · 2023-11-01

**Soundness:** 1 poor
**Presentation:** 2 fair
**Contribution:** 2 fair
**Rating:** 3
**Confidence:** 4

**Summary:**

This paper explores the concept of the verification complexity of Goldstein stationary points. To this end, the authors formulate several problems related to finding (Definition 5), strongly verifying (Definition 6), and weakly verifying (Definition 7) Goldstein stationary points. One direct consequence of these definitions is that the hardness of weak verification implies the hardness of strong verification and finding stationary points (as shown in Lemma 2). The authors demonstrate that efficient deterministic strong verification is achievable by simply applying Caratheodory's theorem. They also argue that weakly verifying a uniform Goldstein stationary point requires an exponential number of queries. Additionally, they claim that if the queries are restricted to a lattice with a width of 10$\epsilon\delta$, then no deterministic algorithm can weakly verify an ($\epsilon, \delta$)-Goldstein stationary point.

**Strengths:**

The perspective of "verification" in this work is interesting, which is fundamentally different from the existing work on "testing" notions of solutions, e.g., the work of Murty and Kabadi (1987), Tian and So (2023), and a missing reference (Yun et al., 2019).  If I understand correctly, an efficiently strongly verifiable function belongs, in some sense, to a function class similar to the complexity class NP. Of course, there are fundamental differences, but the similarity is apparent. From this perspective, the claim made in Lemma 2 is quite natural, suggesting that a problem is not in P if it is not in NP.


Reference:
Yun, Chulhee, Suvrit Sra, and Ali Jadbabaie. "Efficiently testing local optimality and escaping saddles for ReLU networks." International Conference on Learning Representations. 2019.

**Weaknesses:**

## Motivation

My primary concern lies with the motivation behind studying the complexity of the "verification" of Goldstein stationary points. As noted by the authors, an efficient strong "verifier" (as defined in Definition 6) cannot be an efficient "tester" in the conventional sense, as demonstrated by Yun et al. (2019). One possible consideration is to establish lower bounds on finding stationary points by proving the hardness of weak verification, as claimed in this paper. However, this raises several technical concerns.

## Major Technical Points

One such concern is the formal definition of an "omnipotent prover," which I found rather confusing, especially in its usage in Definitions 6 and 7.

- In the proof of Theorem 4, a "computationally unbounded" prover may be insufficient to provide the required vectors {y_i} without additional information about the underlying function f. In contrast, an "omniscient" prover is needed to expose such information. This prover should be able to communicate with the function f to extract the information of $\partial_\delta f$ and return the correct Caratheodory's decomposition {g_i}_i. This is not just a terminology issue but indeed a lack of rigor in the setup of the theoretical framework. This vagueness in definition might become fatal in the following comments.

- The most questionable aspect, in my opinion, is the proof of Corollary 1. As previously discussed, the prover needs to communicate with the underlying function f to facilitate strong verification in Theorem 4. In the proof of Corollary 1, while I understand that $f_0$ and $f_1$ are consistent on the queried points {$x^{(i)}$}, I don't comprehend why an "omniscient" prover would provide exactly the same sequence {$x^{(i)}$} for both functions $f_0$ and $f_1$. As mentioned earlier, a legitimate prover should communicate with both $f_0$ and $f_1$ to provide the most useful information. This raises concerns about the correctness of Corollary 1, which is the main result addressing Q2.

- A similar issue regarding correctness persists in the proof of Corollary 2, as the author states, "the proof is analogous to that of Corollary 1." Corollary 2 is the main result addressing Q3.

## Minor Technical Points

- On page 6, concerning the algorithm, why does the function $f$ need to be differentiable at points {y_i}? Consider the following example: let $\epsilon = 1/2$, $\delta = 1$, and f(x) = max{-1, min{ x, 1 }}. Consider $x=0$. In this case, $y_1 = -1$ and $y_2 = 1$ is the only correct choice, but the function is nondifferentiable at both points.

- A remedy could be replacing the gradients in the Algorithm with Clarke subdifferentials. But this brings a computational issue. How can you ensure the efficient solvability of the convex program in the algorithm? In the general case, even with convexity, this problem could be computationally intractable.

- Another issue is that, in Fact 2, the result is stated for polytopes. But a subdifferential could be far from a polytope. Consider $\partial (x \mapsto ||x||)(0)$.

## Other Comments

- I do not understand why, in Definition 7, completeness and soundness are stated separately and the verifier needs to return a boolean $b \in$ {0,1}. As well-noted by the authors, the "if" condition in the completeness part is always true, so the "verifier" will always return $b=1$, and the "if" condition in the soundness part is also always true.

- In Corollary 2, the required width of the lattice is of the order $\epsilon\delta$, which is too large to have practical implications.

**Questions:**

See comments in Weaknesses.

---

> ### Author Response · Authors · 2023-11-22
>
> We thank the anonymous reviewer for their feedback.
> - We would like to clarify a technical misconception regarding Corollary 1. In particular, we note that the prover need not be honest and may provide adversarial input to the verifier (it is the verifier’s job to decide whether the prover is honest). As such, if we consider $(x^{(i)})$ to be the sequence that the honest prover provides when $f=f_0$, then another (malicious) prover may provide the same sequence to the verifier when $f=f_1$. The verifier needs to provide the same output in both cases (because it has the same input and is deterministic) and so has to fail in at least one of the scenarios above. We will clarify this point as well as our terminology about the prover in future versions of our paper and we thank the reviewer for pointing it out.
> - As the reviewer points out, we need to add a technical clarification for Theorem 4, where the prover has to provide points where $f$ is differentiable. This will incur an arbitrarily small slackness in parameter $\delta$ (between completeness and soundness in Definition 6) and can, otherwise, be done without loss of generality. See also our response to Reviewer 6VnG.
> - Regarding the use of Fact 2, we will add an appropriate clarification for future versions of our paper.
> - Regarding Definition 7, we, once more, emphasize that the prover is not assumed to be honest and the verifier is asked to return the correct value of $b$ regardless of whether the prover is honest or malicious.

---

> > ### Comment · Reviewer_hBbf · 2023-11-22
> >
> > Thank the authors for the response. I have a quick follow-up comment. It seems to me that proving the failure of the verifier for malicious provers is pointless. To demonstrate a lower bound for strong verification, you may need to show the failure of the verifier equipped with the best and honest prover.
> >
> > Besides, it seems to me that certifying the honesty of a prover cannot be achieved by a deterministic verifier within a finite number of oracle calls.

---

> ### Author Response · Authors · 2023-11-22
>
> We respectfully disagree with the reviewer. If the prover is assumed to be honest, then the verification problem is trivial: the prover will provide the correct answer to the verifier who can, in turn, output the prover's response without verifying correctness.
>
> In Theorem 4, we provide a non-trivial algorithm that successfully verifies Goldstein stationarity (according to Definition 6). While the verifier does not directly certify the honesty of the prover, no malicious prover can force the verifier to output a false (positive) answer.
>
> Please note that our Definitions 6, 7 are aligned with standard formulations of the verification problem (e.g., in interactive proof systems).

---

> > ### Comment · Reviewer_hBbf · 2023-11-22
> >
> > Consider the function $f:\mathbb{R}\to\mathbb{R} = t\mapsto max(t,0)$, a point $x=0, \epsilon = 1/2$, and $\delta=1/2$. Let a malicious prover return $y_0 = 1, y_1 = 2$. The correct answer for $x=0$ should be "b=1". But the algorithm (in page 6) return "b=0." I think the crucial point in Definitions 6 (Completeness) is the **existence** of a prover. For a malicious prover, it seems that ensuring the correct answer "b=1" cannot be guaranteed. Please correct me if I missed anything.

---

> > > ### Author Response · Authors · 2023-11-22
> > >
> > > In Theorem 4, we propose a verifier that satisfies Definition 6 for Goldstein stationarity (given access to exact queries and gradient oracles). In Corollary 1, we show that Definition 6 (for uniform Goldstein stationarity) cannot be satisfied by any verifier that only uses a number of zeroth order queries that is less than some exponential threshold.

---

> ### Author Response · Authors · 2023-11-23
>
> To be clear, your example is correct, but it does not contradict any of the claims made in our paper.
>
> Definition 6 states that if $x^*$ is indeed stationary, then some prover can convince the verifier that it is stationary and if $x^*$ is not stationary then no prover can convince the verifier that it is stationary.
>
> In Corollary 1, we show that there is a malicious prover that can convince the verifier that a non-stationary point is stationary (not vice versa).
>
> We hope that this response resolved any additional misconceptions.

---

> > ### Comment · Reviewer_hBbf · 2023-11-23
> >
> > I thank the authors for their responses. The presented results do have potential. I look forward to seeing a better-polished version.

---

### Official Review · Reviewer_FQ87 · 2023-11-01

**Soundness:** 3 good
**Presentation:** 2 fair
**Contribution:** 2 fair
**Rating:** 5
**Confidence:** 3

**Summary:**

This paper studies the theoretical complexity of various problems in non-smooth, non-convex optimization. The basic setup is black-box access to a non-smooth, non-convex function $f$ via either a zeroth-order oracle (i.e. the ability to evaluate $f(x)$) or a first-order oracle (i.e. the ability to compute a generalized gradient $\partial f(x)$). The main tractable solution concept in this setup is a Goldstein stationary point, which is a point $x$ such that there exists a convex combination $g$ of generalized gradients $\partial f(y)$ for $y$ in a $\delta$ neighborhood of $x$, with $\lVert g \rVert < \epsilon$. Notably, from the definition it is not apriori obvious how to even verify whether a given point $x$ is a Goldstein stationary point, unlike in the smooth case where one gradient evaluation suffices.

Prior work (Jordan et al. 2022) gives a randomized algorithm with dimension-independent runtime for approximating a Goldstein stationary point with access to a first-order oracle, and further shows that any deterministic algorithm cannot achieve any convergence without access to both a first and second-order oracle. The same prior work also gives a linear-in-the-dimension lower bound for deterministic algorithms with access to both a first and second order oracle.
The main result of this paper shows that any deterministic algorithm with access to only a zeroth order oracle requires a number of oracle queries that is exponential in the dimension. On the way to this result, the authors study the complexity of verifying that a given point is a Goldstein stationary point, and give an efficient deterministic first-order verification algorithm for this task, under the assumption of arbitrary accuracy of the first-order oracle. The main lower bound of the paper actually applies to the problem of deterministic zeroth-order verification of  Goldstein stationary points, which then straightforwardly implies the same lower bound for any algorithm that finds such a point.

**Strengths:**

Verification of Goldstein stationarity does not seem to have been studied before and it is interesting that one can actually get lower bounds for this problem, which then directly imply lower bounds for the problem of finding stationary points. Furthermore, it is natural to study the verification problem because the definition of Goldstein stationarity for non-smooth functions does not have as obvious of a witness as in the smooth case.

**Weaknesses:**

Given the prior work of Jordan et al, it is not entirely clear what new high-level take-away the theoretical results in this paper provide. The main improvement on problems that were previously studied is the exponential-in-the-dimension lower bound for deterministic algorithms when given access to only a zeroth order oracle. While achieving an exponential lower bound is great, it only holds when one severely limits the class of allowable algorithms (only function evaluation queries, no randomness). In terms of previously studied problems, it would have been more interesting to determine whether the linear-in-dimension lower-bound is tight for deterministic algorithms with access to both first and zeroth order oracles.

The problem of verification of Goldstein points is new to this work and interesting, but the main positive result in this case is a straightforward observation from Caratheodory's theorem, and further utilizes first-order queries of arbitrary accuracy. At this point the most critical open question about verification of Goldstein points appears to be whether one can achieve deterministic verification with finite accuracy. This would demonstrate a nice separation between the problem of verification and finding Goldstein stationary points. However, the main result on query accuracy in the paper is for deterministic zeroth-order algorithms, which demonstrates a dependence on $\delta$ in the accuracy required which is not present for smooth functions. This seems to me like a very minor technical difference, whereas settling the question of whether deterministic verification with finite-accuracy first-order queries is possible seems like it would make a qualitative difference in our understanding of this solution concept.

**Questions:**

Is the use of arbitrary precision necessary for verification of Goldstein stationary points? It is clear that the verification algorithm provided in the paper fails, and it seems plausible that there is some lower bound showing that this is necessary. If it is actually possible with finite precision queries that would also be quite interesting.

---

> ### Author Response · Authors · 2023-11-22
>
> We thank the anonymous reviewer for their useful comments. While we agree that it remains an interesting open question whether the use of arbitrary precision is necessary for verifying Goldstein stationarity via first-order algorithms, we emphasize that our work is the first to formalize the verification problem in this setting.

---

### Official Review · Reviewer_hNC5 · 2023-11-04

**Soundness:** 3 good
**Presentation:** 3 good
**Contribution:** 3 good
**Rating:** 6
**Confidence:** 3

**Summary:**

This paper studies the complexity of deterministic verifiers for nonsmooth nonconvex optimization when interacting with an omnipotent prover. The authors introduce the concept of “weakly verifying Goldstein points”. A lower bound for weak verification of Goldstein stationary points implies a lower bound of finding Goldstein stationary points. In this framework of proving lower bounds, this paper shows that: for deterministic zeroth-order algorithms, both the problem of (1) finding uniform Goldstein stationary points (2) finding (nonuniform) Goldstein stationary points with queries in a lattice are intractable, since the corresponding verification problems are.

**Strengths:**

1. The paper is well-writte. All the concepts and the intuition behind most of the proofs of theorems are clarified clearly.
2. The result of this paper shows that "randomization is necessary (and sufficient) for efficiently finding uniform Goldstein stationary points via zeroth-order algorithms", which is interesting since it forms a sharp contrast with smooth optimization.
3. The lower bounds proved in this article are theoretically solid, and the proofs also look elegant.

**Weaknesses:**

The authors are expected to discuss future work as it has enough space. I believe this could help the reader understand the paper's contributions and the potential impact in future.

**Questions:**

1. The authors claim that they "prove that any deterministic zeroth-order verifier that is restricted to queries in a lattice needs a number of queries that is exponential in the dimension". I wonder what happens in the case of smooth nonconvex optimization. Are there any upper or lower bounds for the queries in a lattice to find an approximate stationary point of a smooth nonconvex function with Lipschitz gradients?
2. The authors say "The main idea behind Theorem 7 is to reduce the construction of f to a geometric question in two dimensions, taking advantage of the symmetry of the lattice". I don't quite understand what this "geometric question" is and what is the "advantage of the symmetry of the lattice". Can the authors explain more about the intuition behind the main idea of the construction in this theorem?

---

> ### Author Response · Authors · 2023-11-22
>
> We wish to thank the reviewer for their constructive feedback. We will add a discussion about open directions in future versions of our paper.
>
> 1. In the smooth setting, there are upper bounds which demonstrate that queries on a lattice with $\mathrm{poly}(\epsilon)$ accuracy are sufficient to find a Clarke stationary point, but we show that in the non-smooth setting, a dependence on $\delta$ is necessary. We refer the reviewer to the discussion below Theorem 3 (pages 3,4).
> 2. Regarding the construction of Theorem 7, the main idea is to project the lattice to a two-dimensional subspace, forming a two-dimensional grid. Then, one can create a high-dimensional function for which the center of the grid is not Goldstein stationary by forming a line that passes through the points of the grid (the line corresponds to the zeros of the function) and ensuring that no angle of the line is too sharp.

---

### Official Review · Reviewer_6VnG · 2023-11-06

**Soundness:** 2 fair
**Presentation:** 3 good
**Contribution:** 2 fair
**Rating:** 5
**Confidence:** 3

**Summary:**

The notion of $(\delta, \epsilon)$-stationarity has emerged as a popular, tractable one for the minimization of nonconvex loss functions prominently featuring in deep learning these days. While several recent works have studied this notion from various angles, this paper explores a fresh question: how can we *verify* Goldstein stationarity? The paper uses connections with Caratheodory's theorem to study this question.

**Strengths:**

Numerous recent papers have studied various questions around $(\delta, \epsilon)$-stationarity: how fast can we get there, can we get there without randomness, how different is this notion from other notions like, e.g., being $\delta$-far from an $\epsilon$-stationary point, etc. This question of verification is **very** nice, in my opinion and definitely deserves to be studied. In my view, this paper's key contribution --- and a big one at that --- is identifying this nice question.

**Weaknesses:**

Please see Q1a and Q1b below. To summarize, I fear that, while formulating a very beautiful question, the paper may not quite satisfactorily be answering the questions.

**Questions:**

Thank you for the interesting and nicely written paper. I have the following main question.

**Question 1a.** Why, in Theorem 4, is it valid to assume that we get a gradient at the query point instead of a subgradient? After all, the function $f$ is assumed to **not** be differentiable everywhere.

**Question 1b.** Following Question 1a, if suppose we were to replace line 2 of the algorithm displayed on Page 6 with $g_i$ being some subgradient of $f$ at $y_i$, then would the algorithm's output of $b=0$ still be correct in all conditions? I fear that this may not be so because a $\delta$-Goldstein subdifferential at $x$ is the convex hull of the union of the subdifferentials in a $\delta$-neighbourhood of $x$, so just because $g$ isn't in the convex hull of the *current* subdifferentials doesn't necessarily mean it doesn't exist in any of the other possible subdifferentials; but then again, it looks like one would need to verify a large number of sets to check this, which seems too costly? I would of course be happy to see a proof showing otherwise! (Also, this question obviously depends on the validity of my Question 1a, so I'd also be happy if this question isn't even valid.)

---

> ### Author Response · Authors · 2023-11-22
>
> We wish to thank the anonymous reviewer for their comments and for alerting us of a missing technical clarification on Theorem 4.
>
> In particular, while it is not explicitly stated in Theorem 4,  the honest prover may without loss of generality only provide points $y_0,y_1,\dots,y_d$ where $f$ is differentiable. This follows from Definitions 1 and 3, where the generalized gradient is defined itself as the convex hull of certain points. These points are limits of convergent sequences of gradient values and each of them can therefore be approximated arbitrarily well by some given point of the sequence. We can therefore define the set $S$ in Fact 1 in terms of points where $f$ is differentiable.
>
> A simpler approach is to assume that $f$ is differentiable, but the gradient is allowed to change arbitrarily quickly within small neighborhoods of points.
>
> We will add this clarification in future versions of the paper.

---

### Meta-Review · Area_Chair_SMzu · 2023-12-06

**Metareview:**

This paper studies the complexity of verifying Goldstein stationary points of non-smooth, non-convex optimization problems by deterministic algorithms that interact with an omnipotent prover. The problem studied is an interesting and important one, as the potential outcomes can shed further light on the computational complexity of non-smooth, non-convex optimization problems. However, the current technical presentation is far from satisfactory. As pointed out by the reviewers, some of the key concepts used, such as "verifier" and "prover", need to be more formally defined, as the author-reviewer discussions did not clarify the matter in a convincing manner. Also, there are still doubts on the technical correctness of the results. The fix proposed by the authors would require another round of thorough review. Furthermore, the technical contributions of the manuscript are limited. Based on the above, I regrettably have to recommend rejection of the manuscript.

**Justification For Why Not Higher Score:**

The exposition of the current manuscript lacks clarity, to the point that it affects the understanding of the technical contributions and raises doubts on the correctness of the technical results. The manuscript has to be significantly revised, which necessitates a thorough re-evaluation.

**Justification For Why Not Lower Score:**

N/A

---

### Decision · Program_Chairs · 2024-01-16

Reject